# The Association between Physical Environment and Externalising Problems in Typically Developing and Neurodiverse Children and Young People: A Narrative Review

**DOI:** 10.3390/ijerph20032549

**Published:** 2023-01-31

**Authors:** Alister Baird, Bridget Candy, Eirini Flouri, Nick Tyler, Angela Hassiotis

**Affiliations:** 1Division of Psychiatry, University College London, London W1T 7BN, UK; 2Institute of Education, Psychology and Human Development, University College London, London WC1H 0AL, UK; 3Department of Civil, Environmental and Geomatic Engineering, Faculty of Engineering Science, University College London, London WC1E 6DE, UK

**Keywords:** physical environment, conduct disorders, intellectual disabilities, aggression, review

## Abstract

The physical environment is of critical importance to child development. Understanding how exposure to physical environmental domains such as greenspace, urbanicity, air pollution or noise affects aggressive behaviours in typical and neurodiverse children is of particular importance given the significant long-term impact of those problems. In this narrative review, we investigated the evidence for domains of the physical environment that may ameliorate or contribute to the display of aggressive behaviours. We have considered a broad range of study designs that include typically developing and neurodiverse children and young people aged 0–18 years. We used the GRADE system to appraise the evidence. Searches were performed in eight databases in July 2020 and updated in June 2022. Additional articles were further identified by hand-searching reference lists of included papers. The protocol for the review was preregistered with PROSPERO. Results: We retrieved 7174 studies of which 67 are included in this review. The studies reported on green space, environmental noise and music, air pollution, meteorological effects, spatial density, urban or rural setting, and interior home elements (e.g., damp/sensory aspects/colour). They all used well validated parent and child reported measures of aggressive behaviour. Most of the studies were rated as having low or unclear risk of bias. As expected, noise, air pollution, urbanicity, spatial density, colour and humidity appeared to increase the display of aggressive behaviours. There was a dearth of studies on the role of the physical environment in neurodiverse children. The studies were heterogeneous and measured a range of aggressive behaviours from symptoms to full syndromes. Greenspace exposure was the most common domain studied but certainty of evidence for the association between environmental exposures and aggression problems in the child or young person was low across all domains. We found a large knowledge gap in the literature concerning neurodiverse children, which suggests that future studies should focus on these children, who are also more likely to experience adverse early life experiences including living in more deprived environments as well as being highly vulnerable to the onset of mental ill health. Such research should also aim to dis-aggregate the underlying aetiological mechanisms for environmental influences on aggression, the results of which may point to pathways for public health interventions and policy development to address inequities that can be relevant to ill health in neurodiverse young people.

## 1. Introduction

The physical environment encompasses all aspects of a child’s physical world and may be defined as objective characteristics of the physical context in which children spend their time (e.g., home, neighbourhood, school). The influence of children’s physical exposures has been summarised differentially by various models, theorems, and theorists over the previous century. Notably, these include the physical environmental elements of children’s exposome (a term introduced by Wild [1,2] regarding the non-genetic influences on outcomes across the lifespan) and Bronfenbrenner’s bioecological model [3,4,5], proposing that children develop within an environmental milieu of five interconnected systems, spanning aspects from urban design (e.g., presence and structure of sidewalks), traffic density, and design of venues for physical activity (e.g., playgrounds, parks, and school yards), to biologically active chemicals, radiation, the internal chemical environment, and psychosocial aspects [6]. The difficulty with these conceptualisations of child development is that they include both physical environmental and (psycho)social influences. As exemplified in a review of the influence of interior hospital environmental interior conditions, Harris [7] segmented the environment into distinct physical exposure categories: ambient, architectural, and interior design.

In this work, an operationalised definition of “physical environment” was incorporated to identify eligible environmental exposures. This classification was derived from a coalescence of Harris’s [7], Bronfenbrenner’s [3,4,5] and Wild’s [1,2] theorems. This resulted in the inclusion of a diverse array of domains, from ambient exposures (sunlight, sound, meteorology), interior design elements (colour, lighting), architectural features (space/spatial crowding), and biological active agents (i.e., air particulate pollutants), to physical aspects of children’s microsystem (i.e., home, school, and neighbourhood characteristics).

A variety of theories have attempted to explain the mechanisms via which environmental domains influence physical and mental health. Although none of these mechanistic models have been fully proven, there are suggestions that positive effects may be the end product of pathways that link several elements, such as mitigation (reduction in air pollution or traffic noise), restoration (stress reduction and attention restoration in alignment with what the Attention Restoration Theory posits) and instoration, whereby attributes of the physical environment, such as greenness in particular, may promote physical activity and social capital and cohesion [8,9,10].

Previous theories have primarily focused on the stress-reducing effects of greenspace, either via a protective influence from harmful environmental stimuli (noise and air pollution) [11,12,13], or via the restoration of attentional resources [14,15]. Recently, it has been posited that greenspaces may provide more direct physiological benefits via increased exposure to phytoncides (plant-derived antimicrobial volatile organic compounds) [16]. Whilst preliminary research into the effects of phytoncide exposure is positive, it is currently inconclusive and additional studies are required [17]. Neuroimaging studies are also shedding insight into potential mechanisms for greenspace exposures potential mechanisms, with one study [18] showing that it can beneficially deactivate the prefrontal cortex in regions linked to depression and rumination.

The literature also indicates that aspects such as ambient air particulate matter exposure may negatively impact development via neuroinflammatory pathways [19,20,21,22,23,24]. Noise pollution may also have detrimental effects via contributions to subjective annoyance and irritation; whilst not necessarily directly causing aggression, noise exposure in those with low threshold for expressing anger may increase its severity [25,26] via draining of attentional and cognitive resources and subsequently leading to increased self-regulatory difficulties [27]. Social-behavioural mechanisms may explain the relationship between behaviour and fluctuations in meteorological effects (such as temperature), e.g., the routine activity theory that proposes that warmer temperatures facilitate more frequent social interaction, increasing opportunity for aggression [28] or that heat increases hostility and physiological arousal and consequently to aggressive behaviour [29]. Theories have posited that high spatial density triggers perceptions of crowding and a subsequent physiological stress arousal response [30,31,32]. Why proximity elicits these responses is still unclear and has been linked to competition for resources and invasion of personal space [33]. Baird et al. [34] reported a beneficial association between household crowding and reduced conduct problems in children with intellectual disabilities. The authors propose several theories about these potentially counterintuitive findings, suggesting that increased availability of and proximity to family members, in intergenerational households, and parental habituation to problematic conduct behaviours are all potential mechanisms underpinning this finding. Using a sensory room unaccompanied may be associated with a sense of autonomy in children and young people which in turn reduces distress [35]. Other pathways may contribute to the impact that music listening has on a broad range of psychological and physiological benefits [36,37,38,39,40,41]. 

As discussed, social-behavioural mechanisms may explain the relationship between aggression and climate effects, for example the routine activity theory proposes that warmer temperatures facilitate more frequent social interaction, increasing opportunity for aggression [28]. Alternatively, the general aggression model (GAM) is more grounded in a physiological aetiology of aggression, suggesting that heat increases hostility and physiological arousal and consequently aggressive behaviours [29].

From the evidence presented so far, it appears that both physical and social environments, in addition to genetic and epigenetic influences, shape the developmental trajectories of children [42,43,44,45,46]. However, in the main, published research is focused on typically developing rather than neurodiverse children [47]. Previous work has evidenced disproportionate influence of children’s early environmental milieu in shaping a range of socio-emotional and cognitive developmental outcomes. Specifically, learning disabled children are more likely to be affected by social adversity, poor housing, and poverty [48]. These children are also exponentially more likely to be exposed to negative environmental exposures such as air pollution [49]. To address failings in supporting these children and their families, an important element is to reduce socioeconomic inequality and improve residential conditions [50]. Furthermore, children with complex neurodisabilities have increased barriers to accessing potentially therapeutic aspects of both the physical and social early environments [51]. Disabled child access to urban greenspaces, for example, is not only infrequent in comparison to their typically developing peers [52], but when significant resources are employed to facilitate access for neurodisabled children, the high-risk nature of visiting these spaces requires rigid structure, impacting on the quality of nature experiences when they do occur [53]. This is one example of the health inequities and disparities experienced by neurodiverse children in comparison to their peers, exemplifying the need for additional research in these domains. 

Externalising disorders are characterised by display of a range of behaviours which are associated with poor impulse-control, and include rule breaking, impulsivity, and inattention; in addition, a core component of these conditions is the presence of heightened aggression. 

Specific child and adolescent externalising disorders include conduct disorder (CD), oppositional defiant disorder (ODD), and attention- deficit-hyperactivity disorder (ADHD). Of particular concern is this repeated presence of aggressive behaviour in these disorders as it is often associated with referral to services and application of a range of restrictive practices, most commonly antipsychotic medications but also inpatient admissions. 

Aggressive behaviours and general behavioural problems such as destructive behaviours have an overall negative influence on carers due to stress and negative interactions between carers and the person they care for, likely resulting in a deterioration of the quality of care [54]. Moreover, behavioural problems are associated with increased service costs because of the impact of behaviours on staff and need for high support levels [55]. Aggressive episodes also provoke concerns about threat to personal safety as well as cause panic and upset [56,57].

These behaviours in both typically developing and neurodiverse children compound societal and educational limitations [58,59,60,61]. They reduce life satisfaction via degradation of social and familial relationships [62], increase economic costs [63], require higher use of physical restraints [64] and restrictive environmental placements [65,66], limit access to support services [67], impair caregiver functioning [68,69], reduce educational opportunities due to teacher burnout [70] and encourage use of restrictive practices including psychotropic medication use [71,72].

Neurodevelopmental disorders (NDDs) are a category of “etiologically diverse conditions” with onset during the developmental period and are characterised by below average intellectual functioning and adaptive behaviour [73]. This classification includes disorders such as intellectual disability (also called learning disability in the UK), autism spectrum disorders (ASD), and other developmental delays (DD). Whilst we appreciate the nuances of the definitions for brevity and clarity, we will refer to those children with NDDs as neurodiversity in this context. 

About one in one hundred individuals has a neurodevelopmental disorder and there are about 351,000 children with intellectual disability in the UK, often coexisting with other neurodevelopmental disorders [74]. Prevalence of aggressive behaviours in NDDs appears to fluctuate depending on sampling methods and assessment strategies, ranging from 8.3% in community samples [75] to 64% in inpatient care [76,77]. Children with intellectual disability were six times more likely to have conduct disorder measured by the Strengths and Difficulties Questionnaire compared to their typically developing peers [78]. Aggressive behaviours are persistent over time [79], with displays of aggression being consistently linked with neurodiversity [57,80,81,82,83,84,85] though prevalence rates reported can be inconsistent. 

Whilst previous research has examined predictors of broadly defined challenging behaviour in children with intellectual disability [86,87,88], none of the studies has included examination of the influence of the physical environment specifically on such behaviours to date. Here, we build on previous work examining the influence of single domains of the physical environment on aggressive behaviour of typically developing and neurodiverse children by including (1) children across the spectrum of ability and (2) all available objective domains of the physical environment.

Therefore, in this narrative review, we examine the certainty of evidence of the impact of the physical environment on typically developing and neurodiverse children’s aggressive behaviours. The outcome of interest was either psychological or biological proxies of aggressive behaviour, annoyance and irritability measured by validated psychometric questionnaires (measures or outcomes which have been empirically evaluated for reliability) or biological markers such as blood pressure, heart rate and skin conductance. The findings are presented by environmental domain (Greenspace, noise pollution, air pollution, meteorology, spatial density, rurality of residence, interior design, and music) and separately for typically developing and neurodiverse children.

## 2. Materials and Methods

### 2.1. Search Strategy

We adhered to the Preferred Reporting Items of Systematic reviews and Meta-analyses (PRISMA) statement checklist [89] in conducting the review, as well as guidance from the Synthesis Without Meta-analysis [90], and the Meta-analysis of Observational Studies in Epidemiology [91] to improve the precision of our reporting. The study protocol was preregistered on PROSPERO (CRD42020160251). Because of the heterogeneity in outcome used and the variation in exposure measures, we were unable to perform a meta-analysis. Instead, we reported the degree of certainty of the evidence available in terms of protective/detrimental, inconclusivity, or no association for each outcome and exposure metric across each domain of the physical environment.

The electronic search strategy comprised 8 bibliographic databases (MEDLINE, PsychINFO, Web of Science, CHINAHLplus, Embase, Cochrane library, EThOS and ProQeust dissertations and theses) and two grey literature sources (NICE evidence search and Google scholar). The inclusion of the latter sources facilitated the retrieval of additional studies from a more diverse range of sources (including policy and public health), whilst mitigating publication bias and increasing the comprehensiveness of the review [92,93]. The search was carried out in July 2020, and replicated in the update to June 2022 with no year of publication limit. Bibliographies of retrieved articles were searched to maximise retrieval of relevant articles. The search strategy was overseen by a specialist librarian (see Appendix A).

### 2.2. Selection Criteria

Studies were included if they (a) reported primary research, (b) were written in English, French, German, Mandarin Chinese and Spanish which were languages spoken by fellow researchers and therefore could be translated, (c) included human participants aged between 0–18 years, (d) contained a psychometrically valid parent or child reported outcome measure of aggressive behaviours or physiological measures of arousal (identified as a proxy measure of aggressive behaviour) and (e) examined exposure to domains of the physical environment.

### 2.3. Screening and Appraisal Process

All retrieved articles were screened by the first author (A.B.). A sub-sample of titles and abstracts (10%) were co-screened by a senior researcher (A.H.) and a post-doctoral researcher (R.R.). Inter-rater reliability for this initial screening was 87% (0.868). Full text data extraction was conducted by the main author (A.B.) using a modified flexible data extraction template used for non-Cochrane reviews [94] with co-screening conducted for a proportion of studies (59%) by independent researchers (see acknowledgments). Substantial agreement between the primary author and co-screeners was reported (83%, κ = 0.6126) with disagreements resolved by the senior researcher (A.H.) who also crosschecked the extraction table for any inconsistencies.

Risk of bias assessment (RoB) and GRADE protocol were adapted from a systematic review by Clark, Crumpler, and Notley [95] on the evidence relating to effects of environmental noise pollution on mental and physical health outcomes. Four items from this review were used to assess the bias for each paper:Evaluation of the quality and validity of the exposure: whether the paper used established or validated environment metrics.Bias due to confounding: whether studies included adjustment for potential confounding variables.Bias due to sampling methodology and reporting of attrition rate.Outcome assessment leading to information bias: whether studies were using validated aggressive behavioural outcome measure(s).

One measure of RoB that was not included in this review was “due to blinding to exposure outcome” as it was not considered appropriate for the methodology of the majority of the retrieved studies which infrequently blind outcome assessors. Overall RoB ratings for each study were aggregates of high, low or unclear across the four domains. We adopted a conservative rating strategy where studies that had equal reports of low and high risk of bias were classified as high.

The GRADE system [96] is a widely used tool recommended by The Cochrane collaboration [97] which provides a ranking of quality for evidence on interventions and relevant outcomes. The modified GRADE approach assigns a priori the highest quality of evidence to longitudinal or intervention studies, and the lowest to cross-sectional studies, subsequently up- or down-grading evidence dependent upon various methodological factors such as RoB, studies not comparing the same variables, inconsistency of findings between studies, imprecision (effect estimate confidence interval containing 25% harm or benefit), publication bias of funnel plot reported, and other considerations (large effect RR > 2, adjustment for all plausible confounding, dose response gradient). As we did not carry out a meta-analysis assessment of GRADE criteria such as precision or publication bias was not possible. 

### 2.4. Measures of Environmental Exposure

Greenspace was measured by land use data percentage of natural space in the neighbourhood (e.g., for the UK, a census output area such as LSOA) or measured within a set distance of the child’s residence. Other indices included satellite derived neighbourhood greenspace (e.g., *normalised difference vegetation index (NDVI)*) and percentage of neighbourhood greenspace.Blue space was measured by parents reporting on number of days taking their children to a beach.Environmental noise pollution included road traffic, construction noise, and aircraft noise.Air pollution was measured by particulate matter, tobacco smoke (nitrogen dioxide: NO2), and elemental carbon attributed to traffic (ECAT).Meteorological variables included seasons, hot or cold weather, humidity and sunlight.Spatial density and interior home/facility design included space per child in square metres (high/low density), wall paint, sensory room, presence of damp.Urbanicity and rurality were described by the location of the child’s residence or school.

### 2.5. Measures of Aggressive Behaviours

The studies utilised a number of psychometrically valid parent and child reported measures of aggressive behaviour, as well as observer ratings. These comprised the full instrument or conduct, aggression, and externalising behaviour domains as follows:Strengths and Difficulties Questionnaire (SDQ) [98]Age-appropriate Behaviour Assessment System for Children, Second Edition (BASC-2) [99]Child Behaviour Checklist (CBCL) [100]Health Related Quality of Life in Children (KINDL-R) [101,102]WHO Global School-based Student Health Survey [103]National Institute of Mental Health Diagnostic Interview Schedule for Children 4th version (NIMH DISC-IV) [104]State-Trait Anger Expression Inventory-2 (STAXI-2) [105]Other outcomes used were observer rated frequency of aggressive behaviour

## 3. Results

The two searches retrieved 7434 records. After deduplication, 7174 were screened of which 257 underwent full-text assessment, resulting in the inclusion of 67 papers (details are shown in the PRISMA flow diagram, Figure 1). Six of which reported on the physical environment and aggressive behaviours in neurodiverse participants.

We report RoB separately for studies carried out with typically developing (Table 1) and neurodiverse populations (Table 2). We follow the same format for the GRADE evidence summaries for the environmental exposures on outcomes of aggressive behaviours for typically developing and neurodiverse children (Table 3 and Table 4).

### 3.1. Typically Developing Children

#### 3.1.1. Greenspace

Eleven longitudinal and seven cross-sectional studies (~46,684 participants) examined associations between greenspace exposure and childhood aggression [34,106,107,108,109,110,111,112,113,114,115,116,117,118,119,120,121,122]. Five studies were carried out in the UK, four in the USA, two in Belgium, with the remaining in Australia, Korea, Lithuania, Germany, Spain and China. All greenspace studies were classified as low RoB.

Inconsistent evidence for harms or benefits was reported across eight studies [106,107,108,109,112,113,117,121] that examined associations between satellite derived neighbourhood greenspace (NDVI) and parental-reported child aggression related outcomes. Two studies [101,109] examining the association between parental-reported child aggression and conduct problems and percentage of land designated as natural land, reported high-quality evidence. Proximity of the child’s residence to greenspace was inconsistently associated with parent reported conduct problems across three studies [106,108,114]. Very low-quality evidence [111,116,120] reported no relationship between percentage of neighbourhood greenspace and both child and parent-reported conduct problems. Moderate-quality evidence from three studies [34,111,120] reported inconsistent beneficial effects of access to private garden space on parent-reported conduct problems.

#### 3.1.2. Environmental Sound and Noise

Three longitudinal and eight cross-sectional studies (*n* = 23,665) assessed the association between environmental noise pollution including road traffic, construction noise, aircraft noise and aggression outcomes [123,124,125,126,127,128,129,130,131,132,133]. These studies were primarily conducted in the UK, Spain, Germany, and the Netherlands, and one study in China. Three of these studies [124,131,132] used data from the multi-national RANCH study examining the influence of high and low road and aircraft noise on the behaviour of pupils who attended schools that were close to main roads or under flypaths. Two studies were judged to be of unclear RoB [126,130], with the majority being rated as low RoB. A very low-quality evidence for harmful association [130] between residential aircraft noise exposure and increased child annoyance was reported. Similarly, low- and very low-quality evidence was found for associations between increased residential noise [126], predicted air and road traffic noise [131], and heightened self-reported child annoyance. Schools located in areas of high aircraft noise were associated with increased child-reported annoyance [127,128,129], but inconsistently correlated with parent-reported child conduct problems ([127,129], both very low quality). Two studies [124,132] examining the role of residential aircraft noise on the parent-administered conduct problems subscale of the SDQ reported no association (low quality). Five studies [123,124,125,133,185] reported very low quality inconsistent evidence for estimated noise exposure effects on parent-reported child aggression. 

Two studies (longitudinal and within-group repeated measures) from the USA (*n* = 658) assessed the association of music on childhood aggression [183,184]. Both studies were rated as high RoB. Aggressive or sexual music content was associated with increased self-reported aggressive behaviour in adolescents ([183], very low quality). Low-quality evidence reported no association between alternating periods of instrumental music and observer rated aggressive behaviours [184]).

#### 3.1.3. Air Pollution

Eight longitudinal and six cross-sectional studies from Lithuania, China, Korea, Iran, Canada, USE, and the UK (*n* = 45,607) explored the influence of air particulate matter on aggressive behavioural outcomes in typically developing children and young people [34,107,116,123,135,136,138,141,142,144,145,146,147,148,149]. One study was rated as high RoB [146], one as unclear RoB [136], and the remaining as low RoB.

Five studies [135,136,146,147] provided either low- or very low-quality evidence supporting the harmful influences of tobacco smoke exposure across various aggressive behavioural questionnaires. Very Low-quality evidence for a harmful association [132] between active or passive tobacco exposure and child self-reported anger and aggressive behaviour was found. 

Three studies [116,141,149] examined the relationship between Nitrogen Dioxide (NO2) exposure and child self-reported conduct problems symptoms and reported inconsistent evidence for a harmful association (Very Low quality). No effect of Elemental Carbon Attributed to Traffic (ECAT) on parent-reported externalising behaviours (BASC-2) was found ([145], Moderate quality). 

In addition, there was inconsistent evidence for an association between exposure to particulate matter less than 2.5 microns (PM2.5), and child self-reported conduct problems [141,149]. No effect was found in a study that examined the influence of PM2.5 on parent reported conduct problem scores [107]. Another study by Loftus et al., 2020 [144] explored the influence of exposure to particulate matter less than 10 microns (PM10) on parent reported child aggressive behaviours but it did not show a significant association (Very Low quality). Ambient air lead exposure (PbA)) was associated with high parent-reported aggressive behaviour ([148], Very Low quality).

#### 3.1.4. Meteorological Exposure

Five longitudinal and two cross-sectional studies (approximately = 6314) from Chile, Canada, the Netherlands, USA, and Italy, assessed associations between meteorological variables and child aggression outcomes [151,152,153,154,155,156,157,158,159]. One study was rated as high RoB [158] with the remaining studies rated as low RoB.

The study by Muñoz-Reyes et al. [158] contrasted the frequency of observed aggressive behaviours during the warm season (summer/spring) with the frequency of such behaviours during the cold season (autumn/winter), reporting Very Low-quality evidence for harmful effect of warm seasonality. Low-quality evidence associated increased humidity with harmful increases in teacher-reported child aggressive behaviours [151,156]. Studies examining the effects of sunlight exposure on teacher reported [151,156] and child self-assessment [152] behavioural outcomes reported inconsistent or no evidence, respectively (very low quality). Low- and Moderate-quality evidence for the harmful influence of increased temperature on teacher and parent-reported child aggression symptoms was reported in three studies [151,156,159]. However, we found one Very Low-quality study that provided evidence for beneficial effect of temperature on children’s self-reported anger [152]. Aggression during summer recess was lower compared with aggression during the school year ([154]; Low quality). No association between hours of precipitation per day and children’s self-reported anger was found ([152], Very Low-quality evidence). Finally, a study carried out by Lochman et al. [153] examined longitudinal associations between tornado exposure and externalising symptoms, and reported a harmful associations of Moderate quality.

#### 3.1.5. Spatial Density and Interior Design

Four observational and two longitudinal studies [34,160,161,162,163,165] (*n* = 8568) from the USA and the UK examined spatial density and architectural design in relation to childhood aggression. RoB was judged as high in all studies except one rated as unclear [163] and one rated as low [34]. A study [165] reported a beneficial effect of increased playroom openness, but no effect of space per child or room group size on observed aggressive behaviours. Low-quality evidence assessing the association between high density (in comparison to low density) child playrooms and frequency of aggressive behaviours reported inconsistent results [160,161,162]. Moderate-quality evidence examining the effect of overcrowding in the home [34,163] reported inconsistent associations with parent-reported conduct problems but was associated with reduced teacher-reported externalising behaviours.

Three studies, two quasi-experimental and one longitudinal (*n* = 8257) conducted in Iran, the UK, and the USA examined the associations between interior design features and childhood aggression [34,179,181]. Low-quality evidence of association ([181] unclear RoB) between red painted classroom walls and increased self-reported aggression was found. In-patient psychiatric ward sensory room modifications were correlated with beneficial reductions in observer rated aggressive behaviour ([179] unclear RoB Moderate quality). Additionally, presence of damp in the house was associated with elevated trajectories of conduct problems in children ([34], Moderate quality).

#### 3.1.6. Urbanicity and Rurality

Three longitudinal and three cross-sectional studies (*n* = 17,630) from the USA, the Netherlands, and Thailand explored the influence of urbanicity and rurality of residence on children’s aggressive behavioural outcomes [34,164,168,170,172,176]. One study was rated as high RoB [168] with the remaining assessed as low or moderate RoB. One study [176] reported inconsistent associations between the location of the participants and scores across three self-reported aggression outcomes (Very Low quality). Moderate-quality inconsistent evidence [34,172] was reported for the effect of urban residence on child conduct problems and aggressive behaviour in parent-reported questionnaires, whilst evidence for a lack of association was found for teacher-completed aggression outcomes [172]. Another study [170] examined the effects of urban or rural settings on aggressive behaviours in schoolchildren attending schools from either setting. It reported no association of setting with parent-reported behaviours, but a harmful effect of urban school location on teacher-assessed behaviours (both Very Low quality). Very Low-quality evidence reported no association between children recruited from rural or urban Head Start centres and teacher-reported anger ratings ([168]: AML Behaviour Rating Scale). Neighbourhood urbanicity (mean number of addresses within a 1 km radius of participant’s residence) was associated with increased teacher-reported child problem behaviours ([166] Low quality). 

### 3.2. Neurodiverse Children

Six studies (*n* = 79,249) from the USA, Pakistan, the UK, and Australia included neurodiverse participants exclusively [34,188,190,191,192,193]. The studies are heterogenous utilising a variety of designs including longitudinal, cross-sectional, quasi-experimental, interventional, including two case studies. Two studies were judged to be of high RoB [190,192], one unclear [193] and the remaining three of low RoB. 

Baird et al. [34] explored interaction effects between a sub-sample of children with intellectual disability (assessed via cognitive measures) from the Millennium Cohort Study (MCS) and various physical environmental exposures (neighbourhood greenspace: NDVI, access to a private garden, air pollution: NO_2_, urban or rural residence, household density, presence of damp. The authors reported no mediating influence of intellectual disability on the association between environmental exposures and children’s conduct problem trajectories, except for household density (beneficial effect, moderate quality). Another study [188] reported no correlation between urban tree canopy coverage and frequency of aggressive behaviours in children with ASD but found an association between residing in lower urban tree canopy areas with increased parent-reported conduct problem severity. However, the evidence was deemed to be of low quality in both studies. The case study by Durand and Mapstone [190] examined the impact of fast and slow beat music on a child with intellectual disability, reporting reductions in observed frequency of aggressive behaviour during the fast beat condition and increases during the slow beat condition in comparison to a no-music baseline (Low quality). Additionally, a clinical trial of new age and classical music [191] provided Moderate-quality evidence for the beneficial effects of music on self-reported aggression in children with intellectual disabilities. A case study [192] assessing the impact of spatial proximity between an adolescent girl with intellectual disability and the therapist, provided a Very Low-quality evidence for a correlation between closer proximity and increased duration of observed aggressive behaviours. Finally, a study [193] examined the efficacy of modified sensory rooms in reducing distress in adolescent psychiatric inpatients, reporting additional benefits for individuals who had a history of aggression (Moderate quality).

## 4. Discussion

This is the first narrative review that updates previous literature across several environmental domains as well as including neurodiverse children, a previously under reported population in other reviews.

### 4.1. Physical Environmental Domains

#### 4.1.1. Greenspace

We found evidence that supports the therapeutic benefits of increased natural land and greenness surrounding child residences. Previous reviews have also shown associations between greenspace exposure and reductions in violent behaviours [194,195]. 

The greenspace evidence synthesised primarily supports the therapeutic influence of neighbourhood nature exposure on child aggressive behavioural outcomes. These effects, at least partially, were also present in NDDs populations. Whilst more epidemiological and experimental research paradigms are required to solidify the evidence for this therapeutic relationship and understand its underlying mechanistic pathways, we provide initial evidence for the role of nature in reducing aggression in neurotypical and diverse children. Initial attempts at establishing guidelines for integration and therapeutic adoption are beginning to be developed [196]. Studies examining socio-cultural barriers to children accessing urban greenspaces [197] are of crucial importance, but these findings need to be communicated to institutions and policy decision makers. We also recommend future experimental studies that aim to elucidate the underlying (neuro)mechanistic pathways via which nature exposure conveys these potential benefits. Advances in this regard would drastically redefine architectural and urban design for physical and mental health. 

#### 4.1.2. Noise Pollution

Children appear to consistently self-report higher aggressive and annoyance related behaviours related to environmental noise, whereas parent reported outcomes either show a lack of association or inconsistent associations both for harm and benefit. This may suggest that noise exposure operates on pathways involving subjective annoyance and irritation which may not translate into objective longer-term increases in aggression problems. Additionally, although noise annoyance may not play a direct role in the aetiology of those problems, noise exposure of individuals who experience frustration or irritable mood has been shown to increase its severity [25,26]. Noise pollution, therefore, may not operate as a causal mechanism of aggression, but exacerbate pre-existing manifestation, potentially via draining of attentional and cognitive resources, leading to increased self-regulatory difficulties [27].

#### 4.1.3. Air Pollution

We found absent and inconsistent associations between ECAT, particulate matter less than PM2.5, particulate matter less than PM10 and NO_2_ exposure and childhood aggression problems. Tobacco smoke exposure showed a harmful association with aggressive behavioural outcomes irrespective of who was the outcome assessor. We also found this harmful association for childhood exposure to ambient air lead exposure (PbA). The lack of association of PM2.5 and PM10 with these behaviours is potentially anomalous when considering research that has linked air pollution with increased risk of mental health disorders [198]. The harmful effects of tobacco smoke and ambient lead exposure may increase the risk of neuropsychiatric disorders and violent crime, possibly via neuroinflammation [19,20,21,22,23,24].

Whilst none of the retrieved articles examined the effects of air pollution on neurodiverse children, it was shown that families of these children disproportionately reside in areas of higher particulate concentration than those of typically developing children [49], as well as exhibiting elevated rates of aggressive behaviour [199,200].

#### 4.1.4. Meteorological Effects

Summer seasonality, humidity, temperature, and previous tornado exposure were consistently correlated with increased childhood aggressive behaviours. We found little evidence for either harmful or beneficial effects of ambient temperature and seasonality. Previous studies suggest that humidity compounds the negative effects of heat on mental health [201], as well as being associated with increased emergency department visits for mental health problems [202]. Elevated temperature has also been associated with increased violent crime [203]; however, those associations warrant further examination.

#### 4.1.5. Spatial Density

The negative impact of high spatial density on aggression in young people [204] and inpatients in psychiatric wards has been highlighted previously [205,206,207]. Notwithstanding the beneficial effects of increased playroom openness, inconsistent influences for other spatial characteristics prevent a firm explanation of findings. Theories have posited that high spatial density triggers perceptions of crowding and a subsequent physiological stress arousal response [30,31,32]. Further studies on possible mechanistic pathways between high spatial density and aggression in children could lead to therapeutic adaptations in clinical and residential spaces [208].

#### 4.1.6. Urbanicity and Rurality of Residence

Due to the quality of retrieved evidence, we were unable to extricate any definitive conclusions for associations between urban or rural residence and childhood aggressive behaviours. This is potentially anomalous considering that children residing in rural areas are exposed to more greenspace which generally appears to have calming effects [209,210,211,212,213,214]. Rurality, however, is only one factor in a great number of confounders on childhood aggression. Furthermore, studies do not often use operationalised definitions of “rural” or “rurality” [215], potentially leading to heterogeneity in the underlying conceptual constructs being examined, limiting the replicability and specificity of results.

#### 4.1.7. Interior Design and Housing Quality

Previous work has associated damp problems with increased toxic mould, contributing to poor air quality [216] and/or potential neuroinflammatory and/or neurotoxic responses [20,22]. Damp in a house may also be associated with other adversities such as low socio-economic status and household disruption [217] exemplified by previous research linking poor household conditions to psychological distress [218].

Whilst preliminary evidence from this review supported the positive impact of modified sensory rooms to de-escalate aggression, it is very limited in scope. One study [35] suggested that the increased reduction in distress related to sensory deficits may be attributable to a sense of autonomy children and young people may gain by using the room unaccompanied.

#### 4.1.8. Music

We found preliminary evidence for the therapeutic potential of music in neurodiverse children which is similar with findings reported in adults [219]. Music listening has been associated with a broad range of psychological and physiological benefits [38,39,40]. Some [36,37] have stated that the therapeutic influences of music may operate mechanistically via enhancing emotional regulation, but such evidence is not yet available [41]. Music is a complex physical phenomenon, which requires additional targeted research to examine its effects on aggressive behaviours in typically developing and neurodiverse young people.

### 4.2. Strengths and Limitations

This review is comprehensive and has examined the evidence of a wide range of environmental exposures in relation to the display of aggressive behaviours in typical and neurodiverse children. To the best of our knowledge, this is the first review that comparatively examines available research on environmental determinants of aggression in these two groups. The review shows clearly the disproportionately sparse literature relating those children and the physical environment despite the fact that they are more likely to be affected by social adversity, poor housing, air pollutants, and poverty [48,49,50,220,221]. 

The incorporation of GRADE to assess the quality of evidence in this review may well be simultaneously both a strength and a limitation. Whilst it facilitated the examination of the certainty of included evidence, the adaptation of GRADE for use in a non-meta-analytic review including epidemiological studies, may, as highlighted previously [95], inadvertently result in downgrading of evidence irrespective of study quality. We also adopted a modified risk of bias protocol which may have impacted the RoB assessments of included studies. There may also have been potential conflicts of interest based on the source of funding which we did not consider in this review.

A final limitation of this evidence synthesis is the inclusion of studies that adopt a diverse range of heterogenous physical environmental exposures and metrics. As has been highlighted previously by experts in physical environmental epidemiological analysis on child socio-cognitive outcomes [116], further research is needed on improving environmental measures of aspects such as air pollution exposure, and access to and quality of children’s greenspaces. Developing more holistic, accurate, and reliable measures of environmental exposures will facilitate novel research paradigms (computational, simulatory and experimental) that can elucidate the influences of these aspects, reciprocally informing direction for future research into (neurobiological)mechanistic pathways. 

## 5. Conclusions

Physical environmental exposures sit at the intersection of social, biochemical, and (epi)genetic aetiological influences on the development and progression of a spectrum of physical and mental health outcomes. Further research can support stakeholders, ranging from city planners and environmental legislators to politicians and clinicians, in considering the role of the physical environment in the context of adverse impact on child (neuro)development. Whilst there is obvious need to further examine environmental and climate influences on mental health of all children, particular attention must be paid to neurodiverse children and their families. A recent report recommended that in order to pursue and achieve health parity for those children, we must “reduce poverty and improve living environments” [50]. Research focusing on that population will help to bridge the equity gap that has significant therapeutic and health implications for all citizens.

## Figures and Tables

**Figure 1 ijerph-20-02549-f001:**
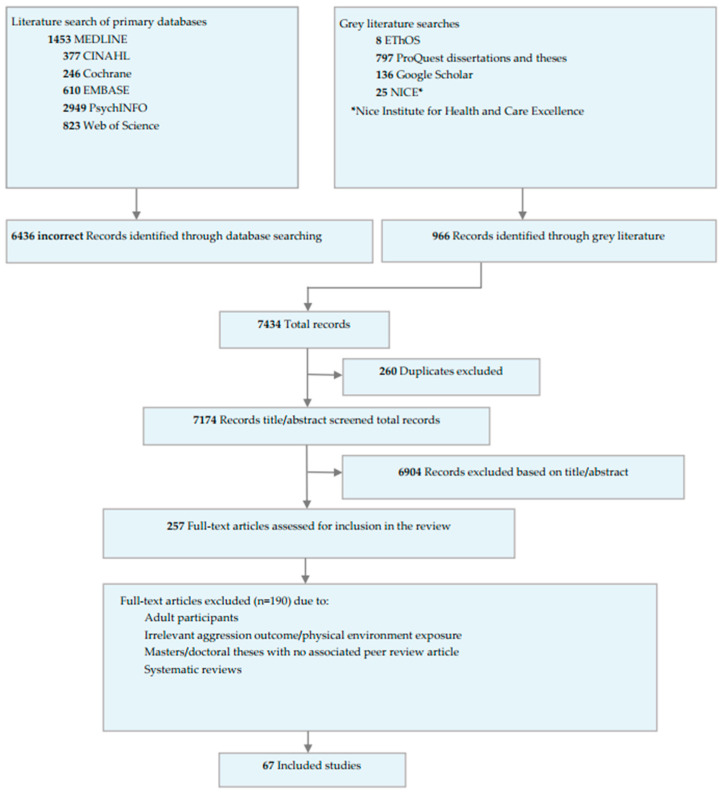
PRISMA flow diagram of the included studies.

**Table 1 ijerph-20-02549-t001:** Risk of Bias (RoB) summary for studies reporting on the association between environmental domains and aggressive behaviours in typically developing children.

Author, Year	Country	Setting	Study Design	*N*	Age (Years)	Follow Up (Years)	Environmental Exposure	Aggression Outcome	RoB	Association
**Greenspace**										
Amoly et al., 2014 [106]	Spain	n.a.	Cross-sectional	2111	7–10	n.a.	**Residential greenspace:** Greenspace surrounding homes in buffer zones measured using the NDVI. **Greenspace playing time:** child’s weekly average time spent playing in greenspaces (hours). **Residential proximity to major green spaces:** if home address was within 300 m of available greenspace. **School greenness and combined home-school greenness:** Average weighted NDVI in a 100 m buffer around school and home locations. **Blue space:** Parental report of how many days they accompanied their children to the beach annually.	Parent-completed Strengths and Difficulties Questionnaire (SDQ) [98] conduct problems subscale.	Low	**Beneficial association:** residential greenspace increased greenspace (NDVI) in 100 m and 250 m buffer radii around child’s residence was significantly associated with decreased conduct problem scores. **No associations:** between greenspace playing time, residential proximity to greenspaces, blue space attendance, and combined school and home greenspace (NDVI) and conduct problems.
Andrusaityte et al., 2020 [107]	Lithuania	n.a.	Cross-sectional	1489	4–6	n.a.	Greenspace (NDVI) in a buffer of 100 m around participants home residence.	Parent-completed Lithuanian version of the Strengths and Difficulties Questionnaire (SDQ) [98].	Low	**No association:** between 100 m NDVI greenspace and risk for conduct problems was reported.
Baird et al., 2022 [34]	UK	n.a.	Longitudinal	8168	3–11	n.a.	**Ward level residential greenspace:** Deciles of the percentage of greenspace within the family’s UK ward. **Access to private garden space:** Parent reported child access to private garden space.	Parent-completed Strengths and Difficulties Questionnaire (SDQ) [98] conduct problems subscale.	Low	**No association:** between ward-level greenspace and conduct problem trajectories over time. **no association** between child access to private garden space and conduct problem trajectories was reported.
Balseviciene et al., [108]	Lithuania	n.a.	Cross-sectional	1468	4–6	n.a.	**Proximity to city parks:** Proximity of residence to nearest park. **Residential greenness:** Greenspace (NDVI) in a buffer of 300 m around participants home residence.	Parent-completed Conduct problems subscale of the Strengths and Difficulties Questionnaire (SDQ) ^3^.	Low	**Beneficial association:** children whose mothers reported low educational attainment reported significantly more conduct problems as the distance of home residence to closest greenspace (parks) increased. **Harmful association:** increased residential greenspace was associated with increased conduct problems in the high maternal education group and approaching significance in the low education group.
Bijnens et al., 2020 [109]	Belgium	n.a.	Longitudinal	7–15	442	n.a.	Seminatural, forested, blue, and urban green areas (green space) in several radius distances (5000, 4000, 3000, 2000, 1000, and 500 m) around residential addresses were calculated	Parent-completed Achenbach Child Behaviour Checklist (CBCL) [100].	Low	**Beneficial association:** for children living in an urban environment, a 1 inter-quartile range increase in greenspace was significantly associated with lower externalising behavioural scores. **No association:** for children residing in rural or suburban areas, no association was reported between greenspace and externalising behaviours.
Feng and Astell-Burt, 2017 [110]	Australia	n.a.	Longitudinal	4968	4	9	Greenspace measured as the percentage of land-use classified as parkland (domestic gardens not included).	Parent-completed externalising behavioural sub-scale (conduct and hyperactivity scale combined) of the Strengths and Difficulties Questionnaire (SDQ) [98]	Low	**Beneficial association:** a non-linear association between increased local greenspace and reductions in children’s SDQ scores was reported, proportional to local land use classified as greenspace.
Flouri et al., 2014 [111]	UK	n.a.	Longitudinal	6384	3	4	Neighbourhood greenspace was defined as the percentage of natural space within groups of census output areas (LSOAs). Private garden access.	Parent-completed conduct problems subscale scores of the Strengths and Difficulties Questionnaire (SDQ) [98]	Low	**No association:** between neighbourhood greenspace and conduct problems was reported. **Beneficial association:** children’s access to a private garden was associated with significantly decreased parent reported SDQ conduct scores.
Jimenez et al., 2021 [112]	USA	n.a.	Longitudinal	908	0–13	13	Greenspace (NDVI) in buffers zones of 90 m, 270 m, and 1230 m centred on participants residence.	Parent and teacher completed externalising subscale of the Strengths and Difficulties Questionnaire (SDQ) [98]	Low	**No association:** persistent exposure to maximum (vs. minimum) greenspace exposure during development was not associated with child externalising behaviours.
Madzia et al., 2019 [113]	USA	n.a.	Longitudinal	313	7	5	Varying spatial buffer zones of greenspace surrounding children’s residence (NDVI) at ages 7 and 12.	Parent-completed externalising subscale scores of the Behavioural Assessment System for Children, Parent Rating Scale, Second Edition (BASC−2) [99]. Scores ≥ 60 classify children as “at risk” for conduct disorder in clinical settings.	Low	**Beneficial association:** greenspace at age 7 was significantly associated with decreased conduct scores at the 200 m buffer radius only. **No association:** no associations were reported between greenspace and conduct scores or aggression scores at age 12 at any buffer radius. **Beneficial association:** increased NDVI at 200 m and 800 m buffers at age 7 was significantly associated with lower probability of being “at risk” of conduct problems.
Markevych et al., 2014 [114]	Germany	n.a.	Longitudinal	1932	9–12	10	Residential proximity to urban greenspace.	Parent-completed conduct problems subscale of the Strengths and Difficulties Questionnaire (SDQ) ^4^.	Low	**No association**.
McEachan et al., 2018 [115]	UK	n.a.	Longitudinal	2594	4.5	4	Varying buffer zones of green space around participants’ home addresses and distance to major green spaces was computed with the normalised difference vegetation index (NDVI).	Parent-completed conduct and hyperactivity subscales (combined to derive externalising behavioural scores) of the Strengths and Difficulties Questionnaire (SDQ) [98].	Low	**No association:** between NDVI and parent-reported externalising behaviours in White British or South Asian participants.
Mueller et al., 2019 [116]	UK	n.a.	Cross-sectional	3683	10–15	n.a.	Greenspace was measured using land use data, which reports percentage of greenspace in the family’s ward (excluding gardens).	Self-completed conduct problems subscale of the Strengths and Difficulties Questionnaire (SDQ) [98].	Low	**No association:** between greenspace and conduct problem scores.
Liao et al., 2020 [117]	China	n.a.	Cross-sectional	6039	5–6	n.a.	Greenspace (NDVI ^1^) in a 100 m buffer zone was measured and weighted assuming that children spent 16 h per-day at home and 8 h at kindergarten.	Parent-completed aggressive behaviour subscale of the Child Behavioural Checklist (CBCL) [100].	Low	**Beneficial association:** increased residence–kindergarten-weighted greenspace was significantly associated with decreased aggressive behaviour scores.
Lee et al., 2019 [118]	Korea	Residence	Cross-sectional	1817	6–18	n.a.	Modified soil-adjusted vegetation index (MSAVI) values were categorised into tertiles (low, moderate, high greenness) and each child was assigned the mean MSAVI within a 1.6 km radius of residence.	Parent-completed externalising subscale (Rule-breaking Behaviour and Aggressive Behaviour combined) of the Child Behavioural Checklist (CBCL) [100].	Low	**Beneficial association:** children residing in highest tertile of average greenness for the 1600 m areas around their homes had significantly lower Externalising Behaviour scores.
Lee and Movassaghi, 2021 [119]	USA	Schools	Cross-sectional	n.a.	5–18	n.a.	Greenspace (NDVI ^1^) in a 100 m buffer zone surrounding schools	Incidence rates of attacks or threats with and without weapons in schools.	Low	**Harmful association:** increased school greenness was associated with increased incidence of threats and attacks (with or without weapons).
Richardson et al., 2017 [120]	UK	n.a.	Longitudinal	2909	4	2	Greenspace defined as the % area of total natural space ^5^ and parks within 500 m of the child’s residence. Private garden access.	Primary caregiver-completed conduct problems subscale of the Strengths and Difficulties Questionnaire (SDQ) [98].	Low	**No association:** between total green space and children’s conduct problem scores. **Beneficial association:** between children not having access to private garden and increased SDQ conduct scores was reported.
Van Aart et al., 2018 [121]	Belgium	n.a.	Longitudinal	172	6–12	6	Semi-natural, forested, and agricultural areas (greenness) and residential and industrial areas in a 5000, 4000, 3000, 2000, 1000, 500, 300 and 100 m buffer from the residential address	Primary caregiver-completed conduct problems subscale of the Strengths and Difficulties Questionnaire (SDQ) [98].	Low	**No association:** between landscape surrounding child’s residence and conduct problems were reported.
Younan et al., 2016 [122]	USA	n.a.	Longitudinal	1287	9	9	Greenspace (NDVI ^1^) was measured in multiple spatial buffers zones for various periods preceding CBCL assessment.	Parent-completed aggressive behaviour subscale of the Child Behaviour Checklist (CBCL) ^2^.	Low	**Beneficial association:** increased greenspace (1000 m NDVI) was associated with significant decreases in aggression.
**Noise pollution**										
Bao et al., 2022 [123]	China	Residence	Longitudinal	3236	7–13	6	Residential road traffic noise was assessed using modelling different periods of the day, including daytime (Lday), nighttime (Lnight), and weighted 24 h (Ldn).	Parental completed conduct problems subscale of the Strengths and Difficulties Questionnaire (SDQ) [98].	Low	**Harmful association:** weighted 24 h (Ldn) noise exposure was associated with increased conduct problems.
Crombie et al., 2011 [124]	UK/Spain/Netherlands	Schools	Cross-sectional	1900	9–10	n.a.	A continuous noise from aircraft and road traffic measure calculated in dB for each school.	Parental completed conduct problems subscale of the Strengths and Difficulties Questionnaire (SDQ) [98].	Low	**No association:** no association between air traffic noise and conduct problems. **Beneficial association:** between increasing road traffic noise and decreasing SDQ problem scores was reported.
Essers et al., 2022 [125]	Spain/Netherlands	Residence	Longitudinal	7958	18 m–9 years	7.5	Average 24 h noise exposure at the participants’ home address during childhood was estimated using EU maps from road traffic noise and total noise (road, aircraft, railway, and industry).	Parental completed Strengths and Difficulties Questionnaire (SDQ) and Child Behavioural Checklist 6–18 (CBCL 6–18) [98,100].	Low	**No association:** between noise exposure and conduct problems or aggressive behaviours was reported.
Grelat et al., 2016 [126]	France	Not reported	Cross-sectional	517	7–11	n.a.	Noise indices were calculated from the front and most exposed façade of the child bedrooms using a noise map.	Child self-report questionnaire on annoyance from various traffic and ambient noise sources.	Unclear	**Harmful association:** increased road and general transport noise exposure was significantly associated with increased child annoyance.
Haines et al., 2001a [127]	UK	Schools	Cross-sectional	340	10	n.a.	Exposure of schools to high and low aircraft noise.	Parent-completed conduct problems subscale of the Strengths and Difficulties Questionnaire (SDQ) [98]. Child self-report questionnaire on annoyance due to; aircraft, train, road and neighbour noise.	Low	**No association:** between aircraft noise exposure at school and SDQ conduct problems was reported. **Harmful association:** increased aircraft noise exposure was significantly associated with increased annoyance.
Haines et al., 2001b [128]	United Kingdom	Schools	Longitudinal	275	8–11	1	Exposure of schools exposed to high or low aircraft noise.	Child self-report questionnaire on aircraft, train, road, and neighbour noise annoyance.	Low	**Harmful association:** higher levels of aircraft noise were associated with significantly elevated levels of annoyance.
Haines et al., 2001c [129]	UK	Schools	Cross-sectional	451	9	n.a.	Exposure of schools to high and low aircraft noise.	Parent-completed conduct problems subscale of the Strengths and Difficulties Questionnaire (SDQ) [98]. Child self-report questionnaire on noise annoyance.	Low	**Harmful association:** increased aircraft noise exposure at school was significantly associated with increased annoyance. **No association:** between aircraft noise and SDQ conduct scores were reported.
Spilski et al., 2019 [130]	Germany	Not reported	Cross-sectional	1243	8	n.a.	Residential aircraft noise over the preceding 12 months (FANOMOS).	Child self-reported annoyance questionnaire.	Unclear	**Harmful association:** between increased aircraft noise and increased child annoyance.
Stansfeld et al., 2005 [131]	UK/Spain/Netherlands	Schools	Cross-sectional	2844	9–10	n.a.	Exposure to external aircraft and road traffic noise was 56,57,58 predicted from noise contour maps, modelling, and on-site measurements.	Child self-report questionnaire on noise annoyance.	Low	**Harmful association:** increased aircraft and road traffic noise was significantly associated with elevated child annoyance.
Stansfeld et al., 2009 [132]	UK/Spain/Netherlands	Schools	Cross-sectional	2844	9–10	n.a.	School exposure to high or low road traffic and aircraft noise.	Parent-completed conduct problems subscale of the Strengths and Difficulties Questionnaire (SDQ) [98].	Low	**No association:** between aircraft noise and conduct problems was reported. **Harmful association:** between increased road traffic noise and higher conduct problem scores.
Tiesler et al., 2013 [133]	Germany	Residence	Cross-sectional	872	10	n.a.	Night (L_night_) and day (L_den_) indicators of road traffic noise at child’s residence were created using weighted long-term annual average sound levels.	Parental completed conduct problems subscale of the Strengths and Difficulties Questionnaire (SDQ) ^4^.	Low	**No association:** between day or night noise exposure and conduct problems was reported.
**Air pollution**										
Andrusaityte et al., 2020 [107]	Lithuania	n.a.	Cross-sectional	1489	4−6	n.a.	Ambient air pollution: Modelled annual mean NO_2_ and PM_2.5_	Parent-completed Lithuanian version of the Strengths and Difficulties Questionnaire (SDQ) [134].	Low	**No association:** between NO_2_ and PM_2.5_ and risk for conduct problems was reported.
Baird et al., 2022 [34]	UK	n.a.	Longitudinal	8168	3–11	n.a.	Annual concentrations of neighbourhood (LSOA) level NO_2_.	Parent-completed conduct problems subscale of the Strengths and Difficulties Questionnaire (SDQ) [98].	Low	**No association:** between NO_2_ exposure and conduct problems was reported.
Bandiera et al., 2011 [135]	USA	Not reported	Cross-sectional	2901	8–15	n.a.	Serum cotinine (a metabolite of nicotine) as a proxy of cigarette smoke exposure.	Parental reported DSM-IV conduct disorder symptoms obtained via The National Institute of Mental Health’s Diagnostic Interview Schedule for Children Version IV (DISC-IV) [104].	Low	**Harmful association:** increased smoke exposure was significantly associated with increased conduct disorder symptoms.
Bao et al., 2022 [123]	China	Residence	Longitudinal	3236	7–13	n.a.	Annual mean concentration of nitrogen dioxide (NO_2_).	Parent-completed conduct problems subscale of the Strengths and Difficulties Questionnaire (SDQ) [98].	Low	**No association:** between annual mean concentration of NO2 and child conduct problem scores.
Bauer et al., 2015 [136]	USA	Community paediatric clinics	Cross-sectional	2441	0–6	n.a.	Self-reported cigarette smoke exposure (screening questionnaire asking families if anyone in the household smoked).	Diagnosis of Disruptive Behaviour Disorder (DBD) was gained from child’s electronic health record. Diagnoses were identified using International Classification of Diseases-ninth revision (ICD-9) [137].	Unclear	**Harmful association:** childhood smoke exposure increased risk of disruptive behaviour disorder.
Gatzke-Kopp et al., 2020 [138]	USA	Residence	Longitudinal	1096	0.5	6.5	Child salivary cotinine (a metabolic of nicotine) was measured as a proxy for exposure to cigarette smoke.	Primary caregiver-completed conduct problems subscale (SDQ) [98] and the Disruptive Behaviours Rating Scale (DBDRS) [139]. These scores were combined to create a composite conduct problems score. Teachers completed the conduct problems subscale (SDQ) [98] and Teacher Observation of Child Adaptation-Revised (TOCA-R) [140].	Low	**Harmful association:** increased cotinine levels associated with increases in a multi-informant latent factor of conduct problems.
Karamanos et al., 2021 [141]	UK	Schools	Longitudinal	4775	11–16	5	Ambient air pollution: Modelled annual mean NO_2_ and PM_2.5_	Child self-report Strengths and Difficulties Questionnaire (SDQ) [100].	Low	**Beneficial association:** NO_2_ and PM_2.5_ were both associated with reduced trajectories of conduct problems over time.
Kelishadi et al., 2015 [142]	Iran	Not reported	Cross-sectional	13,486	6–18	n.a.	Self-reported active, passive, combined or non-smoker status.	Self-reported information on anger and violent behaviours (World Health Organization Global School-based Student Health Survey: WHO-GSHS) [143].	Low	**Harmful association:** increased anger and risk of violent behaviour was associated with any smoker status.
Loftus et al., 2020 [144]	USA	n.a.	Cross-sectional	975	4–6	n.a.	NO_2_ and particulate matter less than 10 microns (PM_10_) at participants’ residences was calculated using a national annual average universal kriging model. Proximity to nearest road.	Parental completed Child Behaviour Checklist (CBCL; ages 1.5–5 years of age) [100].	Low	**Harmful association:** in fully adjusted models, NO_2_ exposure was positively associated with odds of externalising child behaviours. **No association:** was reported for PM_10_ or proximity to nearest road with child externalising behaviours.
Mueller et al., 2019 [116]	UK	Not reported	Cross-sectional	3683	10–15	n.a.	Annual concentrations of neighbourhood level (LSOAs) Nitrogen Dioxide (NO_2_).	Self-completed conduct problems subscale of the strengths and Difficulties Questionnaire (SDQ) [98].	Low	**No association:** between NO_2_ exposure and conduct problem scores were reported.
Newman et al., 2013 [145]	USA	Not reported	Longitudinal	576	1	6	The average daily concentrations of elemental carbon attributed to traffic pollution (ECAT) measured over the child’s first year of life.	Parent-completed aggression and conduct problems subscales from the Behavioural Assessment System for Children, Parent Rating Scale, 2nd Edition (BASC-2) [99].	Low	**No association:** between ECAT exposure and BASC-2 aggression or conduct subscale scores were reported.
Pagani et al., 2017 [146]	Canada	Not reported	Longitudinal	2055	1.5	10.5	Primary caregiver reported household smoking status.	At age 12 children completed questionnaires asking about their antecedent proactive, reactive and conduct problems.	High	**Harmful association:** early exposure to second-hand smoke was significantly associated with increased; conduct problems, proactive, and reactive aggression at age 12.
Park et al., 2020 [147]	Korea	n.a.	Longitudinal	179	5–9	4	Urinary cotinine levels.	Parental completed Korean version of the Child Behaviour Checklist (CBCL).	Low	**Harmful association:** high cotinine levels were significantly associated with increased externalising problems at age 5, but not at ages 7 and 9.
Rasnick et al., 2021 [148]	USA	n.a.	Cross-sectional	263	12	n.a.	Ambient concentrations of lead as a constituent of particulate matter of size 2.5 μm or smaller (PM_2.5_)	Parent-completed Behavioural Assessment System for Children, 2nd edition (BASC-2) [99].	Low	**Harmful association:** birth to 7 months was identified as a sensitive window for lead exposure and aggressive behavioural outcomes.
Roberts et al., 2019 [149]	UK	Home residence	Longitudinal	284	12	6	Exposure to annualised particulate matter less than 2.5 microns (PM_2.5_) and NO_2_ concentrations were estimated at address-level.	Age 12: Conduct disorder symptoms were self-reported and assessed in reference to DSM-IV [150] conduct disorder criteria. Age 18: DSM-IV [150] Conduct disorder diagnoses.	Low	**Harmful association:** increased PM_2.5_ and NO_2_ at age 12 was significantly associated with increased odds for conduct disorders at age 18.
**Meteorological effects**										
Ciucci et al., 2013 [151]	Italy	Day-care centres	Longitudinal	61	2	9 months	Air temperature (°C), relative humidity (%), solar radiation (Jm^−2^) and rain (mm) data.	Teacher-completed Daily Behavioural and Emotional Questionnaire (DBEQ) [151].	Low	**Harmful association:** between increased humidity during winter and increased aggression. **No association:** between other meteorological variables and aggression.
Klimstra et al., 2011 [152]	The Netherlands	Not reported	Cross-sectional	415	Age not reported	n.a.	Sunshine, average temperature, and hours of precipitation.	Self-report anger measured via the Daily Mood Scale ^6^.	Low	**Beneficial association:** between average temperature and anger scores. **No association:** between other meteorological variables and anger.
Lochman et al., 2021 [153]	USA	n.a.	Longitudinal	188	9–13	4	Parent reported tornado exposure measured using the Tornado-Related Traumatic Experiences (TORTE) questionnaire ^7^.	Parent Rating Scale (PRS) of the Behaviour Assessment System for Children (BASC) [99]	Low	**Harmful association:** greater parent reported tornado exposure scores was positively associated with parent reported child externalising behaviours.
Jones and Molano, 2016 [154]	USA	School	Longitudinal	3330	8	2	Development of children during the first two school years, contrasted with scores during summer recess.	Average aggression score from the Teacher Checklist [155].	Low	**Beneficial association:** a significant decrease in aggression was reported during the summer break in comparison to the academic school years.
Lagacé-Séguin and d’Entremont, 2005 [156]	Canada	School	Cross-sectional	33	4	Daily over 30 days.	Humidity, sunshine hours, and temperature (°C).	Externalising behaviours measured via the Teacher-completed Preschool Behaviour Questionnaire (PBQ) [157].	Low	**Harmful association:** increased humidity was significantly correlated with increased externalising behaviours. **Beneficial association:** increased sunshine was significantly correlated with decreased externalising behaviours. **No association:** temperature was not correlated with externalising behaviours.
Munoz-Reyes et al., 2014 [158]	Chile	Schools	Longitudinal	~1000	14–18	1	Cold season (autumn/winter) contrasted to warm season (spring/summer), temperature and humidity.	Observational recordings of school yard aggressive behaviours over an academic year used to construct an aggression intensity index for each participant.	High	**Harmful association:** frequency of aggression was significantly increased during the warm season. **Beneficial association:** increased temperature and humidity were associated with significantly decreased frequency of aggressive events.
Younan et al., 2018 [159]	USA	Residence	Longitudinal	1287	9–10	8	A monthly time-series of average ambient temperature was constructed, and temperature was further aggregated for the periods 1, 2, and 3 years preceding each CBCL assessment.	Parental completed Child Behaviour Checklist (CBCL) [100].	Low	**Harmful association:** between ambient residential temperature 2 and 3 years prior to assessment and externalising behaviours was reported (this effect remained when controlling for urbanicity, humidity, traffic density and proximity to roads or freeways).
**Spatial density**										
Baird et al., 2022 [34]	UK	n.a.	Longitudinal	8168	3–11	n.a.	Household crowding (calculated as the total number of rooms in a residence/total occupants)	Parent-completed Strengths and Difficulties Questionnaire (SDQ) [98] conduct problems subscale.	Low	**Harmful association:** household crowding was significantly positively associated with conduct problems across development.
Ginsburg et al., 1977 [160]	USA	School playground	Observational	28–34	8–11	n.a.	Playground area size (small vs. large).	Observed frequency and duration of aggressive behaviours in the playground.	High	**Harmful association:** smaller play area size was significantly associated with increased physical aggression. **Beneficial association:** duration of aggressive behaviours were significantly shorter in the small play area.
Loo and Kennelly, 1979 [161]	USA	Experimentally designed rooms	Observational	72	5	n.a.	Low-density condition (32.70 ft^2^ per child) and high-density condition (16.35 ft^2^ per child).	Observed frequency of physically aggressive behaviours and anger.	High	**Harmful association:** a significant increase in aggression and anger was reported in the high-density condition.
Loo and Smetana, 1978 [162]	USA	Experimentally designed rooms.	Observational	80	10	n.a.	Low-density condition (52.1 ft^2^ per person) and high-density condition (13.6 ft^2^ per person).	Observed frequency of physically aggressive behaviours and anger.	High	**No association:** between density condition, anger or aggression was observed.
Supplee et al., 2007 [163]	USA	Residence	Longitudinal	120	2	4	Overcrowding of Home (number of rooms divided by total number of people per household).	Maternal completed Child Behaviour Checklist (CBCL) [100] at age 4. Teacher completed Teacher Report Form (TRF) [164] between ages 5.5–6.	Unclear	**No association:** between overcrowding in the home (at age 3) and age 4 maternal reported externalising behaviours. **Harmful association:** overcrowding in the home (at age 3) was significantly associated with increased teacher reported externalising behaviours at age 5.
Neill, 1982 [165]	UK	Nursery	Observational	~100	3–5	n.a.	Playroom openness (POP ratio), space per child and room group size.	Observed frequencies of aggression defined as: “causing distress by any means”.	High	**Beneficial association:** aggression appeared to be higher in more open nursery environments, however due to poor study methodology, associations are unclear.
**Urbanicity and rurality** **(reference category urban where applicable)**										
Baird et al., 2022 [34]	UK	n.a.	Longitudinal	8168	3–11	n.a.	Data from the Office for National Statistics (ONS) was used to assess urbanicity or rurality of children’s residence.	Parent-completed Strengths and Difficulties Questionnaire (SDQ) [98] conduct problems subscale.	Low	**No association:** between geographic location of children’s residence and conduct problem trajectories.
Evans et al., 2018 [166]	The Netherlands	Schools	Longitudinal	895	8–12	4	Neighbourhood urbanicity: mean number of addresses within a circle of 1 km radius around a participant’s residential address.	Teachers completed the Problem Behaviour at School Interview (2000) [167]. Oppositional defiant and conduct disorder subscales were combined into a behavioural problems measure.	Low	**Harmful association:** between urbanicity and behavioural problems was reported (even after full adjustment).
Handal and Hopper, 1985 [168]	USA	Head start and day centres.	Cross-sectional	679	4–5	n.a.	Rural and urban children recruited from Head start centres.	The aggressive subscale of the AML: a behavioural teacher rating tool [169].	High	**No association:** between geography and aggression scores was reported.
Hope and Bierman, 1998 [170]	USA	Residence/schools	Cross-sectional	310	Not reported	n.a.	Schools recruited from urban and rural areas.	Teacher completed Child Behaviour Checklist–Teacher Rating Form (CBCL–TRF) [171]. Parent-completed Child Behaviour Checklist–Parent Rating Form (CBCL–PRF) [171].	Low	**No association:** between parent reported externalising behaviours and children residing in urban and rural environments. **Beneficial association:** urban schoolteachers reported significantly more externalising behaviours than rural teachers.
Sheridan et al., 2014 [172]	USA	Residence	Longitudinal	6550	3	2	Urban or rural classification based on zip code; city, suburban, town, or rural.	Preschool and Kindergarten Behaviour Scales–Second Edition [173]), Social Skills Rating System [174], Family and Child Experiences Survey [175].	Low	**No association:** between geographical location of residence and teacher reported externalising behaviours. **Harmful association:** parents reported higher externalising behaviour in rural children in comparison to city and suburban children but not children residing in towns.
Wongtongkam et al., 2016 [176]	Thailand	Colleges	Cross-sectional	1028	17	n.a.	Participants were recruited from either Rural (Nakhon) or Urban (Bangkok) Thai provinces.	Self-reported violent behaviour: modified from the Pittsburg Youth Study’s measure of serious violence [177]. Violent offences: self-reported via a modified version of the Overt Victimisation subscale of the Problem Behaviour Frequency Scale [178]. Anger expression (internal/external): Frequency of anger was recorded via The State–Trait Anger Expression Inventory–2 (STAXI-2) [105].	Low	**No association:** in self-reported violent behaviour between rural and urban participants was reported. **Beneficial association:** two violent behaviours; “chased with weapons” and “injured someone with weapons” was significantly more frequently reported by urban adolescents. Anger out and in was significantly elevated in the rural condition.
**Interior design**										
Baird et al., 2022 [34]	UK	n.a.	Longitudinal	8168	3–11	n.a.	Parental completed questionnaire on damp problems inside the home.	Parent-completed Strengths and Difficulties Questionnaire (SDQ) [98] conduct problems subscale.	Low	**Harmful association:** parental reported damp problems were significantly associated with increased conduct problems trajectories across development.
Glod et al., 1994 [179]	USA	Inpatient psychiatric ward	Single blind within-groups repeated measures	19 ^8^	10	n.a.	Sensory room modification.	Observer rated aggression using a modified version of the Overt aggression scale [180] in modified and non-modified sensory rooms.	Unclear	**Beneficial association:** aggression after modified sensory room use was significantly decreased in comparison to non-modified room use.
Vakili et al., 2019 [181]	Iran	Classroom	Case-control pre-post design	70	Not reported	12 weeks	Red painted classroom walls vs. a control condition of white walls.	Buss Perry aggression questionnaire [182]	Unclear	**Harmful association:** red classroom walls significantly increased aggression in comparison to the white wall control condition.
**Music**										
Coyne and Padilla-Walker, 2015 [183]	USA	Residence	Longitudinal	548	15	1	Independent assessors rated the physical aggression content of adolescent’s favourite music artists at time 5 (T5).	Self-completed 5-item questionnaire on physical aggressive behaviour at T5 and T6.	High	**Harmful association:** preference for artists with aggressive music content was associated with increased self-report aggressive behaviour.
Hinds, 1980 [184]	USA	Mental health clinic	Within-group repeated measures	10	8–10	n.a.	Alternating 15 min periods of silence and slow Instrumental music.	Observer rated aggressive behaviours.	High	**No association:** between music and no-music conditions in relation to frequency of aggressive behaviour.

^1^ Normalised difference vegetation index (NDVI). ^2^ CBCL Version used: 6–18 years [100]. ^3^ Lithuanian version of the SDQ was used [134]. ^4^ The German version of the Strengths and Difficulties Questionnaire (SDQ) was used in this study [185]. ^5^ The authors use “natural space” and “greenspace” interchangeably. ^6^ An Internet version of the Electronic Mood Device [186] ^7^ Tornado exposure was measured using the Tornado-Related Traumatic Experiences (TORTE) questionnaire [187]. ^8^ Participants were described as comorbid for a “wide range of diagnoses” however no additional information on psychiatric diagnoses was reported.

**Table 2 ijerph-20-02549-t002:** Risk of bias (RoB) summary for studies reporting on environmental domains and aggressive behaviours in neurodiverse children.

Author, Year	Country	Setting	NDD	Study Design	*N*	Age (Years)	Follow Up (Years)	Physical Environmental Measure(s)	Aggression Outcome	RoB	Association
Baird et al., 2022 [34]	UK	n.a.	ID	Longitudinal	155	3–11	8	Neighbourhood greenspace (NDVI), access to a private garden, Air pollution (NO_2_) exposure, Urbanicity of residence, household crowding, household damp exposure.	Parent-completed Strengths and Difficulties Questionnaire (SDQ) [98] conduct problems subscale.	Low	**No association:** between ID diagnosis, conduct problems, and environmental measures (except household crowding) were reported. **Beneficial association:** an interaction effect was reported between ID, home crowding, and conduct problems, reporting that children with ID reported lower conduct problems overtime in more spatially crowded homes.
Barger et al., 2020 [188]	USA	Not reported	ASD	Cross-sectional	70,927	6–17	n.a.	Greenspace (tree canopy percentage).	Frequency of conduct disorder diagnosis and severity of conduct problems via The National Survey of Child Health (NSCH, 2012) [189].	Low	**No association:** % of tree canopy was not associated with diagnosis of conduct disorder. **Beneficial association:** caretakers residing in lower % tree canopy areas, reported more severe conduct problems in children with ASD.
Durand and Mapstone, 1997 [190]	USA	Not reported	ID + Cerebral Palsy + Seizure disorder	Quasi-experimental pre-post design	1	7	n.a.	Non-lyrical fast beat or slow beat music.	Observer rated frequency of challenging behaviours.	High	**Harmful association:** slow beat music was associated with increased challenging behaviours. **Beneficial association:** inversely fast beat music was associated with decreased challenging behaviours.
Gul et al., 2019 [191]	Pakistan	School	ID	RCT	40	6–16	6 months	Music (background new age and classical music).	Child-completed Buss Perry aggression questionnaire total scores [182]	Low	**Beneficial association:** background music was associated with a significant reduction in post-test aggression scores.
Oliver et al., 2001 [192]	UK	Therapy room	ID	Observational case study	1	14	n.a.	Distance of therapist to participant: close (0.067 m) and far distance (2 m).	Mean percentage of time the participant enacted aggressive behaviour.	High	**Harmful association:** increased proximity of the therapist was associated with increased duration of aggressive behaviour.
West et al., 2017 [193]	Australia	Psychiatric inpatient unit	NDDs (not specified)	Pre and post open trial.	112	12–18	Follow up post sensory room use.	Sensory room modification.	History of aggression/The Stepping Stones Sensory Room Questionnaire (SSSRQ) [193] measured distress levels pre- and post-sensory room use.	Unclear	**Beneficial association:** between reductions in client-reported stress following modified sensory room use and history of aggression was reported.

**Table 3 ijerph-20-02549-t003:** GRADE summary of quality of evidence for typically developing children.

Greenspace								
	Satellite Derived Neighbourhood Greenspace (NDVI)		Percentage of Land-Use Classified as Natural	Proximity of Child’s Residence (m) to Nearest GREENSPACE		Percentage of Neighbourhood Greenspace.	Access to Private Garden	Greenness Surrounding Residence (msavi)
Child self-reported aggression and conduct symptoms	n.a.		n.a.	n.a.		Very Low quality ^1^—No effect ^a^ (1)	n.a.	n.a.
Parent reported child aggression and conduct symptoms	Very Low quality ^2^—Inconsistent effect ^b,c,d,e,f,g^ (8)		High quality—Beneficial effect ^h,i^ (2)	Very Low quality ^3^—Inconsistent effect ^j^—(3)		High quality—No effect ^k^ (3)	Moderate quality ^4^—Inconsistent effect ^k^ (3)	Very Low quality ^5^—Beneficial effect ^L^ (1)
() Number of studies included in each GRADE summary is denoted by numeric value inside of parentheses.								
^a^ Mueller et al., 2019 [116]: self-completed Strengths and Difficulties Questionnaire conduct problems subscale (SDQ) [98]. ^b^ Madzia et al., 2019 [113]: Parent-completed externalising subscale scores of the Behavioural Assessment System for Children, Parent Rating Scale, Second Edition (BASC-2) [99]. ^c^ Liao et al., 2020 [111], Younan et al., 2016 [117]: Child Behavioural Checklist (CBCL) [100]. ^d^ Amoly et al., 2014 [106], Balseviciene et al., 2014 [108]: Parent-reported Strengths and Difficulties Questionnaire conduct problems subscale scores (SDQ) [98]/Parental Lithuanian version of the Strengths and Difficulties Questionnaire (SDQ) [134]. ^e^ Andrusaityte et al., 2020 [107]: Parent-completed Lithuanian version of the Strengths and Difficulties Questionnaire (SDQ) [134]. ^f^ Jimenez et al., 2021 [112]: Parent and teacher completed externalising subscale of the Strengths and Difficulties Questionnaire (SDQ) [98]. ^g^ Lee and Movassaghi, 2021 [118]: Incidence rates of attacks or threats with and without weapons in schools ^h^ Feng and Astell-Burt, 2017 [110]: Parent-completed externalising behaviour subscale scores (hyperactivity and conduct subscales combined) of the Strengths and Difficulties Questionnaire (SDQ) [98]. ^i^ Bijnens et al., 2020 [109]: Parent-completed Achenbach Child Behaviour Checklist (CBCL) [100]. ^j^ Markevych et al., 2014 [114], Amoly et al., 2014 [106], Balseviciene et al., 2014 [108]: German parental version of the Strengths and Difficulties Questionnaire (SDQ) [185]), parent-reported Strengths and Difficulties Questionnaire (SDQ) [98] conduct problems subscale scores, parental Lithuanian version of the Strengths and Difficulties Questionnaire (SDQ), [134]. ^k^ Flouri et al., 2014 [111], Richardson et al., 2017 [120], Baird et al., 2022 [34]: Parent-reported Strengths and Difficulties Questionnaire (SDQ) conduct problems subscale scores [98]. ^L^ Lee et al., 2019 [118]: Parent-completed externalising subscale (Rule-breaking Behaviour and Aggressive Behaviour combined) of the Childhood Behavioural Checklist (CBCL) [100].						^1^ Downgraded due to inclusion of cross-sectional study design and inability to assess consistency. ^2^ Downgraded due to inclusion of cross-sectional study design and inconsistency. ^3^ Downgraded due to inclusion of cross-sectional study design and inconsistency. ^4^ Downgraded due to inconsistency. ^5^ Downgraded due to inclusion of cross-sectional study design and inability to assess consistency.		
**Noise pollution**								
	Residential aircraft noise exposure		Residential noise exposure	Predicted aircraft and road traffic noise exposure		High and low aircraft noise exposed schools	Predicted road traffic noise exposure	
Child self-reported aggression and conduct symptoms	Very Low quality ^1^—Harmful effect ^a^ (1)		Low quality ^2^—Harmful effect ^b^ (1)	Very Low quality ^3^—Harmful effect ^c^ (1)		Very Low quality ^4^—Harmful effect ^d^ (3)	n.a.	
Parent reported child aggression and conduct symptoms	Low ^5^ quality—No effect ^e^ (2)		n.a.	n.a.		Very Low quality ^6^—Inconsistent effect ^f^ (2)	Very Low quality ^7^—Inconsistent effect ^e,g,h,i^ (5)	
^a^ Spilski et al., 2019 [130]: Child self-reported annoyance questionnaire (KINDL-R) [101,102] ^b^ Grelat et al., 2016 [126]: Child self-reported questionnaire on annoyance. ^c^ Stansfeld et al., 2005 [131]: Child self-reported questionnaire on annoyance. ^d^ Haines et al., 2001a [127], Haines et al., 2001b [128], Haines et al., 2001c [129]: Child self-report questionnaire on noise annoyance. ^e^ Stansfeld et al., 2009 [132], Crombie et al., 2011 [124]: Parent-completed conduct problems subscale of the Strengths and Difficulties Questionnaire (SDQ) [98]. ^f^ Haines et al., 2001a [127], Haines et al., 2001c [129]: Parent-completed conduct problems subscale of the Strengths and Difficulties Questionnaire (SDQ) [98]. ^g^ Tiesler et al., 2013 [133]: The conduct problems subscale of the German version of the Strengths and Difficulties Questionnaire (SDQ) [185]. ^h^ Bao et al., 2022 [123]: Parent-completed conduct problems subscale of the Strengths and Difficulties Questionnaire (SDQ) [98]. ^i^ Essers et al., 2022 [125]: Parent-completed conduct problems subscale of the Strengths and Difficulties Questionnaire (SDQ) [98] and Child Behavioural Checklist 6–18 (CBCL 6–18) [100].						^1^ Downgraded due to unclear risk of bias and inability to assess consistency. ^2^ Downgraded due inclusion of cross-sectional study design and inability to measure consistency—upgraded 1 level due to large effect size. ^3^ Downgraded due inclusion of cross-sectional study design and inability to measure consistency. ^4^ Downgraded due inclusion of cross-sectional study design and inconsistency. ^5^ Downgraded due inclusion of cross-sectional study design and inability to measure consistency. ^6^ Downgraded due inclusion of cross-sectional study design and inconsistency. ^7^ Downgraded due inclusion of cross-sectional study design and inconsistency.		
**Air pollution**								
	Active or passive tobacco smoke exposure	Nitrogen Dioxide (NO_2_) exposure	Second-hand tobacco smoke exposure	Elemental carbon attributed to traffic (ECAT)	Particulate matter less than 2.5 microns (PM_2.5_) exposure	Particulate matter less than 10 microns (PM_10_) exposure	Ambient lead less than 2.5 microns (PM_2.5_) exposure	
Child self-reported aggression and conduct symptoms	Very Low quality ^1^—Harmful effect ^a^ (1)	Very Low quality ^2^—Inconsistent effect ^b,c,d^ (3)	Very Low quality ^3^—Harmful effect ^e^ (1)	n.a.	Moderate quality ^6^—Inconsistent effect ^c,d^ (2)	n.a.	n.a.	
Parent reported child aggression and conduct symptoms	n.a.	Very Low quality ^9^—Inconsistent effects ^k,l,m,n^ (4)	Low quality ^4^—Harmful effect ^f,g,h^ (3)	Moderate quality ^5^—No effect ^i^ (1)	Very Low quality ^8^—No effect ^k^ (1)	Very Low quality ^10^—No effect ^n^ (1)	Very Low quality ^11^—Harmful effect ^o^ (1)	
Clinician derived aggressive behavioural disorder diagnosis	n.a.	n.a.	Very Low quality ^7^—Harmful effect ^j^ (1)	n.a.	n. a	n.a.	n.a.	
^a^ Kelishadi et al., 2015 [142]: Self-reported information on anger and violent behaviours (World Health Organization Global School-based Student Health Survey: WHO-GSHS) [143]. ^b^ Mueller et al., 2019 [116]: Self-completed conduct problems subscale of the strengths and Difficulties Questionnaire (SDQ) [98]. ^c^ Roberts et al., 2019 [149]: Conduct disorder symptoms were self-reported and assessed in reference to DSM-IV conduct disorder criteria [150]. ^d^ Karamanos et al., 2021 [141]: Child self-reported Strengths and Difficulties Questionnaire (SDQ) [98]. ^e^ Pagani et al., 2017 [146]: Child-completed proactive, reactive and conduct problem questionnaire. ^f^ Bandiera et al., 2011 [135]: Parental reported DSM-IV conduct disorder symptoms via The National Institute of Mental Health’s Diagnostic Interview Schedule for Children Version IV (DISC-IV) [104]. ^g^ Gatzke-Kopp et al., 2020 [138]: Primary caregiver-completed conduct problems subscale of the Strengths and Difficulties Questionnaire (SDQ) [98] and the Disruptive Behaviours Rating Scale (DBDRS) [139], teachers completed the Teacher Observation of Child Adaptation-Revised (TOCA-R) [140]. ^h^ Park et al., 2020 [147]: Parental completed Korean version of the Child Behaviour Checklist (CBCL). ^i^ Newman et al., 2013 [145]: Parent-completed externalising subscale scores of the Behavioural Assessment System for Children, Parent Rating Scale, Second Edition (BASC-2) [99]. ^j^ Bauer et al., 2015 [136]: Diagnosis of Disruptive Behaviour Disorder (DBD) identified using International Classification of Diseases-ninth revision (ICD-9) [137]. ^k^ Andrusaityte et al., 2020: [107] Parent-completed Lithuanian version of the Strengths and Difficulties Questionnaire (SDQ) [134]. ^l^ Baird et al., 2022 [34]: Parent-reported Strengths and Difficulties Questionnaire (SDQ) conduct problems subscale scores [98]. ^m^ Bao et al., 2022 [123]: Parent-reported Strengths and Difficulties Questionnaire (SDQ) conduct problems subscale scores [98]. ^n^ Loftus et al., 2020 [144]: Parental completed Child Behaviour Checklist (CBCL; ages 1.5–5 years of age) [100]. ^o^ Rasnick et al., 2021 [148]: Parent-completed Behavioural Assessment System for Children, 2nd edition (BASC-2) [99].						^1^ Downgraded due to inclusion of cross-sectional study design and inability to assess consistency. ^2^ Downgraded due to inclusion of cross-sectional study design and inconsistency. ^3^ Downgraded due to high risk of bias, indirect physical environmental exposure metric, and inability to assess consistency. ^4^ Downgraded due to inclusion of cross-sectional study design. ^5^ Downgraded due to inability to measure consistency. ^6^ Downgraded due to inconsistency. ^7^ Downgraded due to inclusion of cross-sectional study design and inability to assess consistency. ^8^ Downgraded due to inclusion of cross-sectional study design and inability to assess consistency. ^9^ Downgraded due to inclusion of cross-sectional study design and inconsistency. ^10^ Downgraded due to inclusion of cross-sectional study design and inability to assess consistency. ^11^ Downgraded due to inclusion of cross-sectional study design and inability to assess consistency.		
**Meteorological effects**								
	Summer seasonality	Humidity	Sunlight	Temperature (°C)	Student aggression during summer recess	Hours of precipitation per day	Tornado exposure	
Observer rated child aggression	Very Low quality ^1^—Harmful effect ^a^ (1)	n.a.	n.a.	n.a.	n.a.	n.a.	n.a.	
Teacher reported child aggression and conduct symptoms	n.a.	Low quality ^2^—Harmful effect ^b,c^ (2)	Very Low quality ^3^—Inconsistent effect ^b,c^ (2)	Low quality ^4^—Harmful effect ^b,c^ (2)	Low quality ^5^—Beneficial effect ^d^ (1)	n.a.	n.a.	
Child self-reported aggression and conduct symptoms	n.a.	n.a.	Very Low quality ^6^—No effect ^e^ (1)	Very Low quality ^6^—Beneficial effect ^e^ (1)	n.a.	Very Low quality ^6^—No effect ^e^ (1)	n.a.	
Parent reported child aggression and conduct symptoms	n.a.	n.a.	n.a.	Moderate quality ^7^—Harmful effect ^f^ (1)	n.a.	n.a.	Moderate quality ^8^—Harmful effect ^g^ (1)	
^a^ Munoz-Reyes et al., 2014 [158]: Observational recordings of school yard aggressive behaviours over an academic year used to construct an aggression intensity index. ^b^ Lagacé-Séguin and d’Entremont. 2005 [156]: Teachers completed the Preschool Behaviour Questionnaire (PBQ) [157] ^c^ Ciucci et al., 2013 [151]: Teacher-completed DBEQ questionnaire items derived from the Child Behaviour Checklist (CBCL/2-3) and Early Childhood Behaviour Questionnaire (ECBQ). ^d^ Jones and Molano. 2016 [154]: Teacher Checklist [155]. Beneficial effect of summer recess in comparison to aggression during the school year. ^e^ Klimsta et al., 2011 [152]: Self-report anger measured via the Daily Mood Scale, an Internet version of the Electronic Mood Device [186]. ^f^ Younan et al., 2018 [159]: Aggressive behaviour subscale of the parental completed Child Behaviour Checklist (CBCL) [100]. ^g^ Lochman et al., 2021 [153]: Parent Rating Scale (PRS) of the Behaviour Assessment System for Children (BASC) [99].						^1^ Downgraded due to high risk of bias, indirect measure of physical environment, and inability to assess consistency. ^2^ Downgraded due inclusion of cross-sectional study design. ^3^ Downgraded due inclusion of cross-sectional study design and inconsistency. ^4^ Downgraded due inclusion of cross-sectional study design. ^5^ Downgraded due to indirectness and inability to assess consistency. ^6^ Downgraded due to inclusion of cross-sectional studies and inability to assess consistency. ^7^ Downgraded due to inability to assess consistency. ^8^ Downgraded due to inability to assess consistency.		
**Spatial density**								
	**Increased playroom openness (POP)**		**Space per child**	**Room group size**		**High density playrooms**	**Overcrowding of the home**	
Observer rated child aggression	Low quality ^1^—Beneficial effect ^a^ (1)		Low quality ^2^—No effect ^a^ (1)	Low quality ^3^—No effect ^a^ (1)		Low quality ^4^—Inconsistent effect ^b,c,d^ (3)	n.a.	
Parent reported child aggression and conduct symptoms	n.a.		n.a.	n.a.		n.a.	Moderate quality ^5^—Inconsistent effect ^e,f^ (2)	
Teacher reported child aggression and conduct symptoms	n.a.		n.a.	n.a.		n.a.	Moderate quality ^6^—Beneficial effect ^g^ (1)	
^a^ Neill, 1982 [165]: Observed frequencies of aggressive behaviour. ^b^ Loo and Smetana, 1978 [162]: Observed frequencies of physically aggressive behaviours and anger. ^c^ Loo and Kennelly, 1979 [161]: Observed frequency of physically aggressive behaviours and anger. ^d^ Ginsburg et al., 1977 [160]: Observed frequency and duration of aggressive behaviours in the playground. ^e^ Supplee et al., 2007 [163]: Mother-completed Child Behaviour Checklist (CBCL) [100]. ^f^ Baird et al., 2022 [34]: Parent-reported Strengths and difficulties questionnaire conduct problems subscale scores (SDQ) [98]. ^g^ Supplee et al., 2007 [163]: Teacher completed Teacher Report Form (TRF) [164].						^1^ Downgraded due to high risk of bias and inability to assess consistency. ^2^ Downgraded due to high risk of bias and inability to assess consistency. ^3^ Downgraded due to high risk of bias and inconsistency. ^4^ Downgraded due to high risk of bias and inconsistency of results. ^5^ Downgraded due to inconsistency of results. ^6^ Downgraded due to inability measure consistency.		
**Urbanicity (Reference category urban)**								
	Urban vs. rural residence		Schools recruited from urban and rural areas	Rural and urban children recruited from Head start centres.		Urbanicity (density of residences surrounding address)		
Child self-reported aggression and conduct symptoms	Very Low quality ^1^—Inconsistent effect ^a,b,c^ (1)		n.a.	n.a.		n.a.		
Parent reported child aggression and conduct symptoms	Moderate quality ^2^—Inconsistent effects ^d,e^ (2)		Very Low quality ^3^—No effect ^f^ (1)	n.a.		n.a.		
Teacher reported child aggression and conduct symptoms	Moderate quality ^4^—No effect ^g^ (1)		Very Low quality ^5^—Harmful effect ^h^ (1)	Very Low quality ^6^—No effect ^i^ (1)		Low quality ^7^—Harmful effect ^j^ (1)		
^a^ Wongtongkam et al., 2016 [176]: Questionnaire modified from the Pittsburg Youth Study’s measure of serious violence [177]. ^b^ Wongtongkam et al., 2016 [176]: Frequency of anger was recorded via The State–Trait Anger Expression Inventory–2 (STAXI-2) [105]. ^c^ Wongtongkam et al., 2016 [176]: Self-reported violent offences obtained via a modified version of the Overt Victimisation subscale of the Problem Behaviour Frequency Scale [178]. ^d^ Sheridan et al., 2014 [172]: Parental questionnaire comprised of the: Preschool and Kindergarten Behaviour Scales [173], Social Skills Rating System [174], and Child Experiences Survey [175]. ^e^ Baird et al., 2022 [34]: Parent-reported Strengths and difficulties questionnaire conduct problems subscale scores (SDQ) [98]. ^f^ Hope and Bierman, 1998 [170]: Parent-completed Child Behaviour Checklist–Parent Rating Form (CBCL–PRF) [171]. ^g^ Sheridan et al., 2014 [172]: Teacher questionnaire comprised of: Preschool and Kindergarten Behaviour Scales [173], Social Skills Rating System [174], Child Experiences Survey [175]. ^h^ Hope and Bierman, 1998 [170]: Teacher completed Child Behaviour Checklist–Teacher Rating Form (CBCL–TRF) [171]. ^i^ Handal and Hopper, 1985 [168]: Teacher completed The AML Behaviour Rating Scale [169]. ^j^ Evans et al., 2018 [166]: The Problem Behaviour at School Interview (PBSI) [167].						^1^ Downgraded due to inclusion of cross-sectional study design and inconsistency. ^2^ Downgraded due to inability to assess consistency. ^3^ Downgraded due inclusion of cross-sectional study design and inability to assess consistency. ^4^ Downgraded due to inability to assess consistency. ^5^ Downgraded due inclusion of cross-sectional study design and inability to assess consistency. ^6^ Downgraded due to High risk of bias and inability to assess consistency. ^7^ Downgraded due to High risk of bias and inability to assess consistency.		
**Interior design**								
	Red painted classroom walls		Sensory room modification	Household damp problems				
Child self-reported aggression and conduct symptoms	Low quality ^1^—Harmful effect ^a^ (1)		n.a.	n.a.				
Observer rated child aggression	n.a.		Moderate quality ^2^—Beneficial effect ^b^ (1)	n.a.				
Parent reported child aggression and conduct symptoms	n.a.		n.a.	Moderate quality ^3^—Harmful effect ^c^ (1)				
^a^ Vakili et al., 2019 [181]: Buss Perry aggression questionnaire [182]. ^b^ Glod et al., 1994 [179]: Observer rated aggression using a modified version of the Overt aggression scale [180]. ^c^ Baird et al., 2022 [34]: Parent-reported Strengths and Difficulties Questionnaire (SDQ) conduct problems subscale scores [98]).						^1^ Downgraded due to High risk of bias and inability to assess consistency. ^2^ Downgraded due inability to assess consistency. ^3^ Downgraded due inability to assess consistency.		
**Music**								
	Alternating 15 m periods of silence and Instrumental music.		Aggressive content of child’s favourite music artists					
Observer rated child aggression	Low quality ^1^—No effect ^a^ (1)		n.a.					
Child self-reported aggression and conduct symptoms	n.a.		Very Low quality ^2^—Harmful effect ^b^ (1)					
^a^ Hinds, 1980 [184]: Observer rated aggressive behaviours. ^b^ Coyne and Padilla-Walker, 2015 [183]: Self-completed 5-item questionnaire on physical aggressive behaviour.						^1^ Downgraded due to high risk of bias and inability to assess consistency.^2^ Downgraded due to high risk of bias, indirect measure of physical environment, and inability to assess consistency.		

**Table 4 ijerph-20-02549-t004:** GRADE summary of quality of evidence for neurodiverse populations.

Physical Environmental Exposure(s)								
	Satellite Derived Neighbourhood Tree Canopy Percentage	Non-Lyrical Fast Beat Music	Non-Lyrical Slow Beat Music	New Age and Classical Music	High Proximity of Therapist (Compared with Low Proximity)	Sensory Room Modification	Greenspace (NDVI), Air Pollution (NO_2_), Private Garden Access, Urban Residence, Household Damp Problems.	Household Crowding
Clinician derived aggressive behaviour disorder	Very Low quality ^1^—No effect ^a^ (1)	n.a.	n.a.	n.a.	n.a.	n.a.	n.a.	n.a.
Parent reported child aggression and conduct symptoms	Very Low quality ^1^—Beneficial effect ^b^ (1)	n.a.	n.a.	n.a.	n.a.	n.a.	Moderate quality ^6^—No effect ^g^ (1)	Moderate quality ^7^—Beneficial effect ^g^ (1)
Observer rated child aggression	n.a.	Low quality ^2^—Beneficial effect ^c^ (1)	Low quality ^2^—Harmful effect ^c^ (1)	n.a.	Very Low quality ^3^—Harmful effect ^d^ (1)	n.a.	n.a.	n.a.
Child self-reported aggression and conduct symptoms	n.a.	n.a.	n.a.	Moderate quality ^4^—Beneficial effect ^e^ (1)	n.a	Moderate quality ^5^—Beneficial effect ^f^ (1)	n.a.	n.a.
() Number of studies included in each GRADE summary is denoted by numeric value inside of parentheses.								
^a^ Barger et al., 2020 [188]: Previous clinician diagnosed aggressive behavioural disorder such as: Oppositional Defiant Disorder or Conduct Disorder retrieved from The National Survey of Children’s Health (NSCH) [189]. ^b^ Barger et al., 2020 [188]: Parental reported severity of child’s conduct problems retrieved from The National Survey of Children’s Health (NSCH) [189]. ^c^ Durand and Mapstone, 1997 [190]: Observer rated frequency of challenging behaviours. ^d^ Oliver et al., 2001 [192]: Mean percentage of time participant enacted aggressive behaviour. ^e^ Gul et al., 2019 [191]: Child-completed Buss Perry aggression questionnaire total scores [182]. ^f^ West et al., 2017 [193]: History of aggression and distress via The Stepping Stones Sensory Room Questionnaire (SSSRQ) [193]. ^g^ Baird et al., 2022 [34]: Parent-reported Strengths and Difficulties Questionnaire (SDQ) conduct problems subscale scores [98]).						^1^ Downgraded due to inclusion of cross-sectional study design and inability to assess consistency. ^2^ Downgraded due to high risk of bias and inability to assess consistency. ^3^ Downgraded due to High risk of bias, Indirectness, and inability to assess consistency. ^4^ Downgraded due to inability to assess consistency. ^5^ Downgraded due to inability to assess consistency. ^6^ Downgraded due to inability to assess consistency. ^7^ Downgraded due to inability to assess consistency.		

## Data Availability

Not applicable.

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
