# Peer review of "The Association between Physical Environment and Externalising Problems in Typically Developing and Neurodiverse Children and Young People: A Narrative Review"

_ijerph, 2023, doi:10.3390/ijerph20032549_

Round 1

Reviewer 1 Report

Comments by Reviewer 1

               The relationship between the physical environment and externalizing behavior disorders has received considerable attention over the years, but there are few systematic and integrative reviews of this literature. Hence a well conducted systematic review of this scattered literature would be a welcome addition. This review fills that gap. The review is reported clearly and follows the best standards of systematic literature reviews. In addition, it takes a relatively large number of varied papers and imposes useful order on them by analyzing them into type of environmental variable, amount of evidence, quality of evidence, and effect. Tables 3 and 4 (once you get your head around the large quantity of data) tell a useful story that is well presented. The upshot of all this work is that there are gaps in the literature, much of it is of poor quality, the relationship between environmental variables are ambiguous and, where consistent, weak and probably of small effect size, mediated by numerous unknown variables and largely do not address children with neurodevelopmental disorders: Good to know! So, I recommend that some version of this paper is published.

               I think this paper might advance the field in a number of ways. First, the independent variables measured here are often measured in a crude and somewhat arbitrary fashion. How should we measure exposure to green spaces? Size of the back yard, back yard yes / no, number of minutes running around in the backyard or various ways of splitting exposure to the backyard into high / low, hi / medium / low levels etc? These choices, which are made in such different ways could easily determine if a relationship between the IC and DV are detected and why studies come up with different results.

               A second issue is that almost all of the identified studies are correlational. This the independent variable – say person density in a household – might index social facts like irritating encounters, or a host of factors relating to income. Correlational studies cannot disentangle these confounds. One thing this paper might do is to highlight the few experimental papers that do indeed answer the question of mechanism and causality and, if they find no relationship between the independent and dependent variable, suggest how future research might address this. There are more than a handful of experiments out there on environmental manipulation and criminal aggression that illustrate this approach.

               A final issue to consider is that these average results probably hide large individuals differences between individual differences. What is it about one child who has high exposure to high levels of environmental noise who behaves aggressively die to that noise and another who does not?

               In sum, this is a good paper that accurately summarizes and analyzes a messy but interesting literature and raises more questions than that literature can yet answer. I recommend publication, and suggest the authors and editor consider these points in a revision if they wish to do so.

               It was a pleasure to read this well conducted study.

Author Response

Thank you for your comprehensive review of this work, we appreciate all your comments and have attempted to improve the manuscript based on these recommendations, please see attached doc to see responses. 

Reviewer 2 Report

This clearly written manuscript describes a thorough and methodologically-sound review on an important topic. Specific comments and suggested revisions are listed below.

In general, I was surprised that the review found so few studies on the impact of physical environments on neurodiverse children. I assume this is because the review focussed on externalising behaviour outcomes but this should be acknowledged in the introduction and/or discussion. There are some gaps in the coverage of the current literature on the relationship between physical environment and neurodevelopmental disorders (see for example, this review which includes some of the same studies as this manuscript: https://www.sciencedirect.com/science/article/pii/S0048969722007008)

Abstract:

Line 28 - "...although the certainty of the association was low." It's not clear what association this refers to until further down the abstract when the statement is repeated. Consider removing this part of the sentence.

Lines 36-37 "more likely to experience adverse early life experiences including living in more deprived environments". I'd like to see this expanded on in the introduction. Is there evidence to suggest that neurodiverse children are more likely to be exposed to specific environmental domains that might impact their health. (e.g. have less access to greenspace), if not this should be highlighted as another gap in the literature, in the introduction.

Introduction:

Lines 46-50 - please provide a reference this definition, in part to justify expansion of domains.

Lines 50 & 51 - I'm not clear on the difference between weather conditions and meteorological effects.

Lines 77-80 & 95-97: repeated information/text.

Lines 101-103: what specific developmental trajectories does evidence suggest are affected? Does previous literature focus on specific behaviours (e.g. aggression) rather than a broader range of externalising behaviours?

Lines 108-118: two forms of externalising behaviour (reactive and proactive) are described in depth but it's unclear how this distinction relates to the rest of the manuscript - is there evidence that one form or the other are impacted differently by physical environment? The review does not further consider these different types of externalising behavour. Consider removing this section.

Line 169: please briefly list the specific environmental domains here. Also see line 198: were any domains excluded? how was this operationalised?

Line 196: " psychometrically valid..." how was this operationalised/ assesed by the reviewers?

Methods: the methodology suggests that the review considered broad externalising behaviours, however the results suggest a focus on aggression.  Was this an a priori criterion or was there a paucity of studies on other types of behaviours? Does the review consider that all externalising problem behaviours are aggressive? In my view these are different constructs, and aggression is one component of externalising problem behaviours. The switch in terminologies is confusing but likely easily remedied with clarification in the introduction (and methodology).

Line 272: "The two searchers...". Should be "searches"?

Figure 1: article excluded due to "Irrelevant aggression outcome..." Should be "Irrelevant externalising behaviour outcome?" See also Tables 1 & 2 "Aggression outcome" columns and comment above.

Line 274: "Sixty-one of which reported on the physical environment and externalising behaviours in neurodiverse participants". Should be "Six..." (61 included neurotypical children and six include neurodiverse children).

Tables 1-4: consider putting all footnotes after the whole table, putting them after each section makes the tables hard to read and results in repetition. Further, in Tables 3 & 4, there is a lot of repetition in the footnotes: information that has already been included in Tables 1 & 2, plus the same reasons for downgrading listed separately.

Discussion

Line 539: "the fact that they are more likely to be affected by social adversity, poor housing and poverty". This statement needs a reference. I would also like to see further discussion on the evidence (or lack of) for whether neurodiverse children have different exposures to specific physical environment domains.

The abstract concludes that: "research should also aim to dis-aggregate the mechanisms of action for both environmental influences on the externalising problems". But, this is only mentioned in the Discussion in relation to spatial density and there is no further discussion on how this might be done, more generally.

Author Response

Thank you for your review, it is very thorough and has helped us to improve the manuscript, please see the attachment for a segmented response to comments. 
